# Constraints on axion-like dark matter from a SERF comagnetometer

Itay M. Bloch [1,2], Roy Shaham [3,4], Yonit Hochberg[5], Eric Kuflik[5], Tomer Volansky[6] & Or Katz [7,8] ✉

Ultralight axion-like particles are well-motivated relics that might compose the cosmological dark matter and source anomalous time-dependent magnetic fields. We report on terrestrial bounds from the Noble And Alkali Spin Detectors for Ultralight Coherent darK matter (NASDUCK) collaboration on the coupling of axion-like particles to neutrons and protons. The detector uses nuclei of noble-gas and alkali-metal atoms and operates in the Spin-Exchange Relaxation-Free (SERF) regime, achieving high sensitivity to axion-like dark matter fields. Conducting a month-long search, we cover the mass range of $1.4 \times 10^{-12}$ eV/$c^2$ to $2 \times 10^{-10}$ eV/$c^2$ and provide limits which supersede robust astrophysical bounds, and improve upon previous terrestrial constraints by over two orders of magnitude for many masses within this range for protons, and up to two orders of magnitude for neutrons. These are the sole reliable terrestrial bounds reported on the coupling of protons with axion-like dark matter, covering an unexplored terrain in its parameter space.

It has been known for nearly a century that most of the matter in our universe is non-luminous. The presence of this so called Dark Matter (DM) has been confirmed by a wide range of observations on all scales. However, to date, all observations rely on the DM's gravitational interactions, precluding the determination of its particle identity, including its mass and interactions. A highly motivated and well-studied class of DM candidates are Axion-Like Particles (ALPs)[1–3] in the ultralight mass regime[4,5]. Originally introduced to solve the strong CP problem of the Standard Model of particle physics[6–8], the axion and its generalizations can explain the observed DM relic abundance, and at the same time, predict feeble non-gravitational interactions with visible matter which can be probed experimentally[4].

ALPs may interact with gluons or photons, or with fermions such as protons, neutrons and electrons. Various experiments explore constraints on these possible couplings of the ALP DM[9–11]. Most classical direct detection experiments search for DM from masses at the eV/$c^2$ scale and above via absorption by or scattering from target materials[12]. While ultralight ALPs cannot be detected using such techniques, significant progress has been made recently in the search for them using experiments on earth[13–24]. For light enough ALPs, the predicted density is high enough so that it can be treated as a classical background field which may be detectable by other means. In particular, when interacting with fermions, the ALP exhibits itself as a coherent narrow-bandwidth time-dependent, anomalous field which interacts with the spins of target fermions[25].

Atomic magnetometers, Nuclear Magnetic Resonance (NMR) sensors and co-magnetometers are terrestrial detectors which are utilized for the search of ALPs-sourced anomalous fields and other signatures of physics beyond the Standard Model[14,15,20–22,24,26–33]. These sensors use spin-polarized atoms in a gaseous, liquid, or solid phase which collectively respond to the anomalous-magnetic field of the ALP and can thus detect or constrain the couplings of ALP to fermions. Typically, the mass range in these searches is set by the relaxation rate of the spins and their resonance frequencies, which determine the range in which the spins are most sensitive to oscillating fields.

[1]Berkeley Center for Theoretical Physics, University of California, Berkeley, CA 94720, USA. [2]Theory Group, Lawrence Berkeley National Laboratory, Berkeley, CA 94720, USA. [3]Rafael Ltd., 31021 Haifa, Israel. [4]Department of Physics of Complex Systems, Weizmann Institute of Science, Rehovot 76100, Israel. [5]Racah Institute of Physics, Hebrew University of Jerusalem, Jerusalem 91904, Israel. [6]Department of Physics, Tel Aviv University, Tel Aviv, Israel. [7]Duke Quantum Center, Duke University, Durham, NC 27701, USA. [8]School of Applied and Engineering Physics, Cornell University, Ithaca, NY 14853, USA. ✉ e-mail: or.katz@cornell.edu

Previous searches have utilized dual atomic species to search for ALP DM at small masses, usually using nuclear spins as at least one of the species[15,22,24,26,34,35]. They relied on simultaneous overlapping of the resonance frequencies of the two species at the oscillating frequency of the ALP, which limited the measurement bandwidth to $f \lesssim 1$ Hz (about $4 \times 10^{-15}$ eV/$h$). Owing to the limited search duration (typically up to several months), previous bounds were cast on $m_{DM} \lesssim 4 \times 10^{-12}$ eV/$c^2$[14,34,36]. To constrain higher ALP masses one may focus on the response of ALPs at a single broadband resonance and operate in the SERF regime[37-39] to suppress non-magnetic noise sources. While several techniques have been proposed previously[40], to this day none has been directly implemented above $m_{DM} > 4 \times 10^{-14}$ eV/$c^2$[34].

Furthermore, the most stringent terrestrial constraints of the coupling between ALPs and fermions in the ultra-light mass regime are based on atomic detectors that use nuclear spins of noble gases. These spins are composed predominantly by their valence neutrons, which enables measurable ALP-neutron interactions. However, cases where the ALP-proton interaction dominates are also theoretically motivated[41]. For instance, in models of hadronic axions (e.g. the KSVZ model[42,43]), ALPs do not couple to quarks and leptons at high energies, and as a consequence their coupling to neutrons is small or could possibly vanish, while their coupling to protons remains sizeable[41]. Constraining the coupling between ALPs and protons requires nuclear spins whose proton contribution is large and known to a sufficient accuracy. Therefore to this day, no reliable bounds on ALP DM interaction with protons were reported in any terrestrial technique.

Here we report on experimental constraints on ALP DM interactions with protons and neutrons. The results rely on a month long search using a dual-specie spin ensembles of polarized $^3$He and $^{39}$K (potassium) atoms. The former is sensitive to ALP-neutron coupling and the latter to ALP-proton coupling, utilizing the nonzero neutron or proton contributions to the spin in their nuclei. Operating the detector in the Spin-Exchange Relaxation-Free (SERF) regime with a high number-density for the potassium, we suppress the effect of photon shot noise and improve on the current terrestrial limits of the coupling to neutrons (protons) by as much as two (three) orders of magnitude in the mass range $1.4 \times 10^{-12} - 2 \times 10^{-10}$ eV/$c^2$. Moreover, barring the

uncertain supernova constraints, the ALP-proton bound improves on all existing terrestrial and astrophysical limits, partially closing the regions for couplings in the range $5 \times 10^{-6}$ GeV$^{-1}$ to $2 \times 10^{-5}$ GeV$^{-1}$ and in the range $(1-9) \times 10^{-3}$ GeV$^{-1}$.

## Results

### ALP Interactions with Fermions

We use a gaseous mixture of alkali-metal and noble-gas spins whose nuclear spins can couple to the ALPs as shown in Fig. 1. The interaction Hamiltonian of ALP fields with noble-gas spins is given by

$$H_{ALP-He} = g_N \mathcal{A} \cdot \mathbf{N} \qquad (1)$$

where $\mathbf{N} = \sum_n \mathbf{N}_n$ is the collective nuclear spin operators of the noble-gas ensemble, summing over the operators of all noble-gas spins in the measurement volume. The vector $\mathcal{A} = \nabla a$ denotes the gradient of the ALP field $a$, which is a stochastic variable whose spectral content depends on the energy density and the velocity distribution of DM $\mathbf{v}_{DM}$[44], and is concentrated in a narrow band of frequencies near $\omega_{DM} = (m_{DM}c^2 + \langle \mathbf{v}_{DM}^2 \rangle/2)/\hbar$. The factor $g_N$ represents the coupling between ALPs and the particles that compose the nuclear spin. For $^3$He (spin 1/2), the dominant contribution comes from its single valence neutron, such that $g_N = \epsilon_N g_{aNN}$. We take $\epsilon_N \approx 0.85$ for the fractional contribution of neutron spin to the nuclear spin of $^3$He atoms[45]. $g_{aNN}$ is the ALP-neutron coupling coefficient we aim to bound.

We use spins of alkali-metal atoms to detect the Helium response as an optical SERF magnetometer, and concurrently, to constrain the coupling between ALPs and protons as illustrated in Fig. 1b. $^{39}$K atoms have a single valence electron (spin-1/2) as well as a nonzero nuclear spin. The interaction Hamiltonian of ALP fields with alkali-metal nuclei is given by

$$H_{ALP-K} = g_I \mathcal{A} \cdot \mathbf{I}. \qquad (2)$$

Here $\mathbf{I} = \sum_i \mathbf{I}_i$ denotes the collective nuclear spin operator of the alkali-metal atoms, summing over all alkali-metal atoms in the

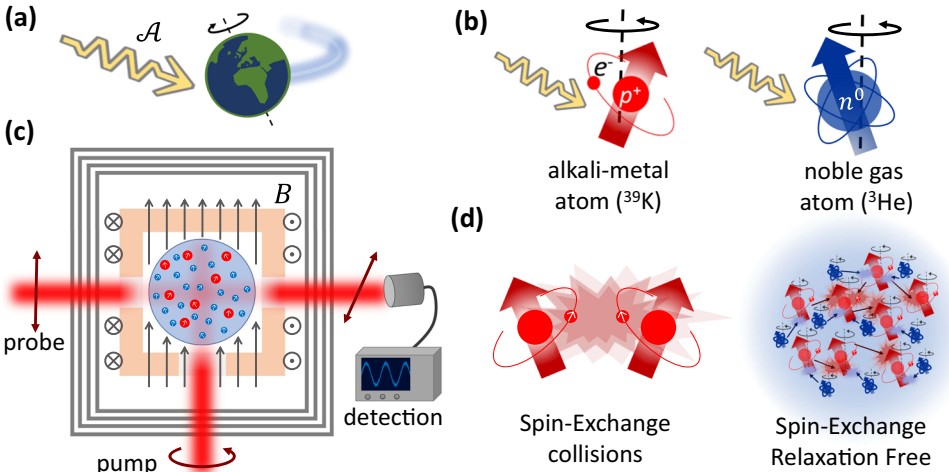

**Fig. 1 | Search for ALP dark matter using a SERF comagnetometer. a** As the Earth moves through the dark matter halo, matter on Earth can potentially interact with the Axion-Like Particles (ALPs) dark matter in a non-gravitational manner. Such interactions can manifest through the gradient field $\mathcal{A} = \nabla a$ of the ALP coupling to other particles. **b** ALPs coupling to atoms. The coupling to alkali-metal and noble-gas nuclear spins is manifested as an anomalous-magnetic field which drives the precession of the spins. The coupling to noble-gas spins is predominantly through their valence neutron ($n^0$) whereas the coupling to alkali-metal nuclei is predominantly through their proton hole in their nucleus[46] ($p^+$, which is drawn with an orbiting electron, $e^-$). **c** Detector configuration. Spin-polarized $^{39}$K (red) and $^3$He (blue) gases are contained in a spherical glass cell, and their precession at a constant magnetic field $B$ is optically monitored. The magnetic field determines the resonance precession frequency of the atoms and thus also the ALP masses for which the detector is sensitive. **d** Operation in SERF regime. Operation with high potassium density renders random spin-exchange collisions between pairs of $^{39}$K atoms very frequent, but also suppresses their relaxation (Spin-Exchange Relaxation-Free). Collisions between $^3$He and $^{39}$K coherently couple the spin gases and enables to superpose the signal of the $^3$He on the optically measured $^{39}$K.

measurement volume. We use natural abundant potassium atoms (predominantly $^{39}$K with spin-3/2) which have a proton hole that dominates the nuclear spin[46]. We then cast $g_I = \epsilon_P g_{aPP}$ where $\epsilon_P \approx 0.16 \pm 0.04$ is the fractional contribution of proton spin to the nuclear spin for $^{39}$K where the uncertainty denotes the spread of prediction by the models reviewed in ref. 46. These models also report on uncertain and much smaller fractional contribution of neutron spin, which is consistent with or is exactly zero; in this search we neglect this contribution. $g_{aPP}$ is the ALP-proton coupling coefficient we aim to bound.

## Apparatus

The detector is comprised of a naturally abundant potassium vapor and 1500 Torr of $^3$He gas that are enclosed in a spherical glass cell of 1.4cm diameter as shown in Fig. 1c. The cell also contains 40 Torr of N$_2$ gas to mitigate radiation trapping and assist the optical pumping process. The two spin ensembles are initially unpolarized; we orient the spins by continuous optical-pumping of the potassium spins, reaching a polarization degree of $P_K \approx 85\%$. Subsequently, the helium spins are polarized by collisions with the optically-pumped potassium atoms, in a process known as spin-exchange optical-pumping (SEOP)[47], reaching a polarization degree of $P_{He} \approx 20\%$. The spins are aligned along an externally applied magnetic field $B_z \hat{z}$ and are magnetically shielded from the ambient field. The magnetic field together with applied light-shifts and the spin-exchange shifts generated by the two polarized spin gases determine the electron paramagnetic resonance (EPR) frequency $\omega_K$ of the potassium spins and the nuclear magnetic resonance (NMR) frequency $\omega_{He}$ of the helium spins. We use an off-resonant optical probe to measure the collective total spin of the alkali-metal atoms along the $\hat{x}$ direction via polarimetry technique, using a pair of photo-diodes (Thorlabs PDB210A) in a homodyne configuration[48], with each of the two photo-diodes receiving about 0.5mW of power (estimated using the Faraday rotation of the polarization during calibration measurements).

We operate the detector in the SERF regime, for which relaxation by spin-exchange collisions between alkali-metal pairs and also by spin-destruction collisions is highly suppressed[39,49–52]. This regime is realized by using an elevated potassium density for which the EPR frequencies we consider are much slower than the rapid rate of spin-exchange collisions $R_{SE} \approx 5 \times 10^5$ s$^{-1}$. It allows us to use a large number of potassium atoms $N_K \approx 5 \times 10^{14}$ to increase the apparatus sensitivity, yet maintain a low decoherence rate; e.g. at $f_K = 10$ kHz, the measured magnetic decoherence rate (which is the total decoherence rate including power broadening by continuous optical-pumping, residual spin-exchange relaxation and other relaxation processes) is $\Gamma_K = 770$ Hz, about three orders of magnitude smaller than $R_{SE}$. The sensitivity of the apparatus to magnetic or anomalous fields oscillating near the EPR resonance is $\sim (1-3)\,\mathrm{fT}/\sqrt{\mathrm{Hz}}$ between $2-32$ kHz, but at most search frequencies, the sensitivity to anomalous fields is dominated by a magnetic-field noise floor which is below $9\,\mathrm{fT}/\sqrt{\mathrm{Hz}}$ (see Methods and SI). Below 2 kHz, a reduced magnetic sensitivity and the lack of near resonant magnetic measurements hinder the decomposition of the noise to its magnetic and non-magnetic contributions (see Methods and SI).

## Response to ALP field

We can describe the response of the collective spins of the alkali-metal and noble-gas spins to the ALP field with the Bloch-equations. A weak oscillatory field in the $xy$ plane $\mathcal{A}_+ = \mathcal{A}_x + i\mathcal{A}_y$ exerts a torque that rotates the orientation of the collective spins of the two gases off the $\hat{z}$ axis, and generates a steady precession. The response of the spins depends on the amplitude of the ALP field and on the frequency difference of $\omega_{DM}$ from their magnetic resonance. Anomalous fields that couple to neutrons tilt the helium spins and produce an oscillating spin

component with amplitude

$$\langle N_+ \rangle = \frac{P_{He} N_{He}}{2} \frac{g_N \mathcal{A}_+}{\omega_{He} - \omega_{DM} - i\Gamma_{He}}. \qquad (3)$$

Here $\langle N_+ \rangle = \langle N_x \rangle + i\langle N_y \rangle$ denotes the mean collective spin of the helium ensemble in the transverse direction and $\Gamma_{He} < 0.1$ Hz is the spin decoherence rate. In our search range $\omega_{DM} \gg \omega_{He} \gg \Gamma_{He}$.

Owing to the great separation between the EPR and NMR resonances considered in this search ($\omega_{DM}, \omega_K \gg \omega_{He}$) the dynamics of the two spin gases is weakly-coupled[53–56]. Consequently, the total collective spin of the potassium $\langle F \rangle = \sum_i (\langle I_i \rangle + \langle S_i \rangle)$, comprised of summation over the potassiums' nuclear and electron spins, simultaneously responds to the torque exerted by the precessing helium and by its direct coupling to the ALP. Representing the collective spin of the alkali-metal vapor in a complex form $\langle F_+ \rangle = \langle F_x \rangle + i\langle F_y \rangle$, we derive the total spin response to ALP fields

$$\langle F_+ \rangle = \frac{P_K N_K}{2} \frac{(\zeta g_{aPP} - \xi g_{aNN})\mathcal{A}_+}{\omega_K - \omega_{DM} - i\Gamma_K}, \qquad (4)$$

where the unitless parameter $\xi$ depends inversely on $\omega_{DM}$ and varies in the search between 46 at low frequencies to 0.29 at high frequencies and $\zeta = 0.69$ (see Methods). The potassium spin component $\langle F_x \rangle$ is directly detected by the optical probe, allowing to simultaneously measure $g_{aPP}$ and $g_{aNN}$.

## Data acquisition

We searched for ALP fields with $f_{DM}$ in the range of $0.33 - 50$kHz (corresponding to a mass range of 1.4–200 peV/$c^2$) by recording the precession of the potassium in the absence of any applied oscillating fields. We varied the magnetic field at few tens of discrete points, and at each point recorded the detector's response for a duration that is longer than the coherence time of the ALP and is about $10^7 - 10^8$ oscillations of the EPR frequency at that field, see Table S2 for the details for several specific measurements. Over a 1-month period, we completed 5 different scans with different samplings of the magnetic field, with three of them used for limit-setting. These three had magnetic fields in the range $B_z = 1 - 7\mu$T to gain sensitivity for ALP fields that oscillate in the reported search range. Each measurement was preceded by initial pumping of the helium spins at $B_z = 7\mu$T, and was preceded and appended by a calibration of the magnetic response and the helium spin polarization. The first and fourth scans were designated to optimize the analysis procedure and were not used to cast the main bounds, as was decided before unblinding. The values of the magnetic field points that were used in the search, along with the magnetic calibration data, are presented in the SI and in[57].

## Search results

We use the log-likelihood ratio test to constrain the presence of ALP DM with 95% confidence level (C.L.) bounds, presented in Fig. 2 for the ALP-neutron coupling (left) and ALP-proton coupling (right). The stochastic nature of the ALPs was treated using the method of ref. 14 (see also refs. 13,28,58 for similar procedures). These constraints cover the mass range between 1.4 peV/$c^2$ and 200 peV/$c^2$, measured with a resolution slightly higher than the line-width of the ALP wavepacket (about $3 \times 10^{-7} m_a$ taking $m_a$ as the ALP mass). Out of ~5 × 10$^6$ spectrally-distinct ALP candidates (this number is estimated by binning the studied frequency range with a typical ALP bandwidth of about $10^{-6} f_{DM}$), we have identified a few thousand spectral points that are inconsistent with our simple white noise model and appear as spikes in Fig. 2. While the search only aims to exclude ALPs and does not attempt at a possible discovery, we have decided to conduct additional analyses post-unblinding, including a comparison of the spectral shape of the observed spikes with the expected ALP signal. Our analysis reveals

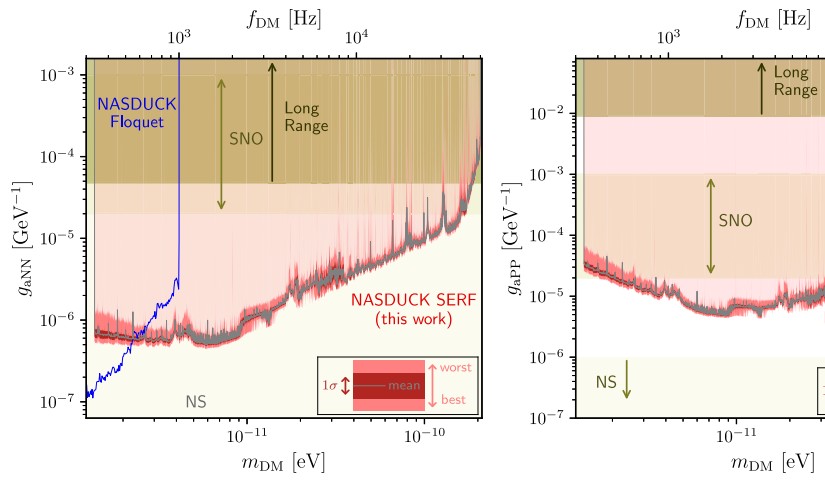

**Fig. 2 | Constraints on ALP-neutron and ALP-proton couplings.** In this work we use a Noble and Alkali Spin Detectors for Ultralight Coherent darK-matter (NAS-DUCK) collaboration Spin-Exchange-Relaxation-Free (SERF) comagnetometer to constrain Axion-Like-Particle (ALP) Dark Matter (DM) couplings over a broad range of ALP masses $m_{DM}$ (with $f_{DM} \equiv m_{DM}c^2/h$). We present 95% C.L. limits. The light transparent red region shows the exclusion region for the ALP-neutron couplings $g_{aNN}$ (left) and ALP-proton couplings $g_{aPP}$ (right). Due to the finite resolution of the figure and given the dense set of measurements, the limits appear as a bright red band. The width of this band denotes the strongest and weakest values around each mass point. The precise tabulated bounds can be found in ref. 57. The black solid line shows a binned average of the bound while its $1\sigma$ variation is shown as the red

band (both calculated in log-log space at a binning resolution of 0.1% of the mass). Other terrestrial constraints from searches for long-range forces (which do not search for dark matter)[27,59] (olive-green region) and from NASDUCK-Floquet[14] (blue-line) are presented. In beige we depict the excluded regions from solar ALPs unobserved at the Solar Neutrino Observatory (SNO)[60], and in light beige the stellar constraints from neutron stars cooling (NS)[61–65] (see text for further discussion of the exact range of validity of this bound for $g_{aPP}$). We do not show the model-dependent limits from supernova (SN) cooling considerations and neutrino flux measurements[4,66,67], which would constrain the entire range presented in the figure, since they notably rely on the unknown SN collapse mechanism[68].

that, with the exception of a few spikes, most of the observed data is unlikely to be explained by ALP candidates. For a detailed discussion of this process and a description of the remaining spikes, please refer to the SI.

While the entire bounds as a function of mass is tabulated in ref. 57, due to finite image resolution, we illustrate the limits by their geometric mean in log-spaced bins of size 0.1% of the mass (gray lines), surrounded by the standard deviation (computed in log-space) spread of bounds in the same bins (red), further surrounded by the minimal and maximal bounds found in each bin (bright red). All couplings above the bright red bands are excluded (light transparent red region).

The olive green regions show constraints from other long-range searches on ALP-neutron[27] and ALP-proton[59] couplings (which do not search for dark matter). The blue line is the mean exclusion line of the NASDUCK-Floquet experiment[14]. Our results provide complementary probes to stellar constraints. The beige regions come from searches for ALP-emission from the sun by the Solar Neutrino Observatory (SNO)[60] and from neutron star (NS) cooling considerations (light beige)[61–65]. The constraints on ALPs from stellar systems, including those from SNO and NS, are unable to exclude ALPs with large couplings due to them becoming trapped at the stars. The exact point at which these bounds stop is hard to determine with exactness. We take the upper edges of the NS bounds to be $g_{aNN} \lesssim 10^{-4}$ GeV$^{-1}$ based on ref. 61, and $g_{aPP} \lesssim 10^{-6}$ GeV$^{-1}$ based on Ref. 62. We note that the two upper edges of these constraints are rough estimates that were calculated independently by other groups[61,62] and carry a relatively large difference that may be an artifact of different approximations used, rather than some underlying difference in the physics. The upper edges of the constraints on both ALP interactions from SNO were taken from ref. 60. For both neutron and proton couplings, the shown regions are constrained by model-dependent supernova (SN) cooling considerations or neutrino flux measurements from SN1987A[4,66,67]. Since the theoretical reliability of such limits is arguable due to the unknown SN collapse mechanism[68], we do not plot them here. The derived limits of this search, dubbed NASDUCK-SERF, on the ALP couplings to neutrons, improve the existing terrestrial limits over

nearly the entire mass range by up to two orders of magnitude, providing a strong complementary probe to stellar constraints from NS and SN. Furthermore, our bound on ALP-protons interactions in a large part of the mass range from 2 peV/$c^2$ to 100 peV/$c^2$ lies in regions that are otherwise only excluded by model-dependent arguable SN constraints.

## Discussion

It is interesting to compare the operation and sensitivity of our detector with self-compensating co-magnetometers that hold the strongest terrestrial bounds on neutron coupling at low frequencies[15,69,70]. Self-compensating co-magnetometers use similar mixtures of alkali-metal and noble-gas spins to detect anomalous fields but set the EPR frequency $\omega_K$ near resonance with the NMR frequency $\omega_{He}$ to realize damped and near-critically coupled dynamics. This operation enhances the signal to noise ratio for detecting oscillating anomalous fields that act on the noble-gas spins at very low frequencies below $f_{DM} \lesssim 10$ Hz, by suppressing magnetic noises[71]. However, for the range of ALP frequencies that we consider in this work, $\omega_{DM} \gg \omega_{He}$ such that the low-frequency enhancement and suppression mechanisms are rendered ineffective, and neither the alkali-metal nor the noble-gas spins are resonant with the driving field. Our detector instead, sets the EPR resonance near the frequency of the searched ALP, and by that fully exploits the high sensitivity of the SERF magnetometer. This operation therefore improves the sensitivity to anomalous fields by about a factor of $\omega_{DM}/\Gamma_K$ compared to self-compensating co-magnetometers, which is >100-fold at the high-frequency end of our search range.

Various cosmological scenarios can give a variety of ALP couplings that would produce the observed cold dark matter density[72]. However, the range of ALP-neutron or ALP-proton couplings predicted are typically several orders of magnitude smaller than our current bounds (when comparing ALP-nucleon couplings to ALP-photon couplings such as those discussed in ref. 72, it is important to remember that often the former is enhanced by three orders of magnitude when compared to the latter due to an expected $\alpha_{EM}/2\pi \approx 10^{-3}$ suppression

on the ALP-photon coupling, with $\alpha_{EM}$ the fine-structure constant). Nevertheless, sensors based on ensembles of alkali-metal and noble-gas spins have the potential to explore this theoretically motivated region in the ALP parameter space, thanks to their long coherence time and the size of the ensemble[73]. In addition to setting constraints, our work represents progress towards experimental exploration of this theoretically motivated regime by leveraging the high sensitivity of SERF sensors for this frequency range to search for new physics.

Several techniques are expected to improve the performance of our sensor. The constraints set by our search for most spectral points are predominantly limited by the magnetic field noise that is produced by our innermost magnetic shield layer[74]. Replacement of this layer by a low-noise material (e.g. MnZn feritte[75]) is expected to reduce the magnetic noise level by a factor of $\mathcal{O}(5)$. Furthermore, it is possible to improve the bounds on neutrons by using a denser noble-gas and hybrid SEOP, which are expected to further improve the noble-gas spin magnetization. In future measurements, it would be advantageous to conduct independent control measurements of magnetic fields to enhance the discovery capability of ALP-spin interactions. Magnetic field gradients can be used to depolarize the nuclear spins, which enables measurement of the background in ALP-neutron interaction searches and distinguishes magnetic noise sources from ALP candidates. Alternatively, a second magnetometer based on a different technology (e.g.[76]) can be used to provide an in-situ measurement of the magnetic background that couples differently to ALPs. Even if the sensitivity of the second magnetometer is lower, it can be used to suppress narrow peaks observed in the data to the sensitivity level of these detectors and greatly improve the constraints on ALPs in regions currently dominated by these spurious peaks.

## Methods
### Spin dynamics in the SERF regime
The electron and nuclear spins of the potassium are coupled via the strong hyperfine interaction $H_{hpf} = A_{hpf}\mathbf{I}_i \cdot \mathbf{S}_i$, which sets the total spin in the electronic ground-state $\mathbf{F}_i = \mathbf{I}_i + \mathbf{S}_i$ of the $i$th atom as an operator with good quantum numbers. We can describe the dynamics of the collective spin of the potassium ensemble $\mathbf{F} = \sum_i \mathbf{F}_i$ and the helium spin ensemble $\mathbf{N} = \sum_i \mathbf{N}_i$ in the $xy$ plane using the coupled Bloch equations[77]

$$\frac{d}{dt}\langle\mathbf{F}\rangle = -\left(\gamma_e\mathbf{B} + \frac{2qA_{He}}{N_{He}}\langle\mathbf{N}\rangle\right) \times \langle\mathbf{S}\rangle - g_I\mathcal{A} \times \langle\mathbf{I}\rangle - \Gamma_K\langle\mathbf{F}\rangle, \quad (5)$$

$$\frac{d}{dt}\langle\mathbf{N}\rangle = -\left(\gamma_N\mathbf{B} + \frac{2A_K}{N_K}\langle\mathbf{S}\rangle\right) \times \langle\mathbf{N}\rangle - g_N\mathcal{A} \times \langle\mathbf{N}\rangle - \Gamma_{He}\langle\mathbf{N}\rangle. \quad (6)$$

Here $\gamma_e$, $\gamma_N$ are the gyro-magnetic ratios of the bare electron and helium spins respectively, and $\mathbf{B} = B_z\hat{\mathbf{z}}$ is the magnetic field. $q(P_K)$ is the slowing-down-factor[78], monotonically decreasing from $q(P_K = 0) = 6$ to $q(P_K = 1) = 4$ as a function of $P_K$, the potassium degree of polarization. It is experimentally determined by measurement of the gyro-magnetic ratio of the potassium in our setup, giving $q = 4.3$. $A_{He}$ ($A_K$) are the total spin-exchange shifts of the magnetic levels had all helium (potassium) spins were completely polarized, i.e. $|\langle\mathbf{N}\rangle| = N_{He}/2$ ($|\langle\mathbf{S}\rangle| = N_{He}/2$), and in our experiment $A_{He} \approx 18$ kHz and $A_K \approx 0.8$ Hz. We denote the collective spins of the electrons and nuclei of alkali-metal atoms $\langle\mathbf{S}\rangle = \sum_i\langle\mathbf{S}_i\rangle$ and $\langle\mathbf{I}\rangle = \sum_i\langle\mathbf{I}_i\rangle$. We denote by $\Gamma_{He}$ and $\Gamma_K$ the transverse relaxation rates of the helium and potassium spins respectively.

The ALP fields interacting with the alkali-metal nuclei exert a torque on the collective nuclear spin $\langle\mathbf{I}\rangle = \sum_i\langle\mathbf{I}_i\rangle$. Here we consider a dense alkali ensemble in the SERF regime, for which rapid spin-exchange collisions between pairs of alkali-metal atoms drive the spin state to follow a spin temperature distribution, and correlate the mean collective spins by $\langle\mathbf{F}\rangle = q\langle\mathbf{S}\rangle = \frac{q}{q-1}\langle\mathbf{I}\rangle$[78]. The number of polarized atoms are $P_K N_K = 2|\langle\mathbf{S}\rangle|$ for the potassium and $P_{He}N_{He} = 2\langle\mathbf{N}\rangle$ for the helium[77,79].

In our experimental setup, the spins are nearly aligned with the $z$ direction, such that $|\langle\mathbf{S}\rangle| \approx |\langle S_z\rangle|$ and $|\langle\mathbf{N}\rangle| \approx |\langle N_z\rangle|$.

In the regime we operate the apparatus, the EPR frequency associated with the precession of the total alkali-metal spin $\omega_K = |\frac{1}{q}B\gamma_e + P_{He}A_{He}|$ is about two orders of magnitude larger than the NMR frequency of the helium spins $\omega_{He} = |\gamma_N B + P_K A_K|$. This different scaling originates from the difference in the gyromagnetic ratios, and the application of magnetic field that is aligned with spin-exchange field produced by the helium ($A_{He}$), in contrast with the operation of self compensating co-magnetometers which use an inverted magnetic field[56,69]. Considering the spectral response of the helium spins to oscillating dark matter fields that are near the resonance of the potassium spins, we obtain Eq. (3) as the fourier transform of Eq. (6) after neglecting the small effect of $\langle S_+\rangle$ on the noble-gas dynamics.

The response of the transverse collective spin $\langle F_+\rangle$ can similarly be represented in the frequency domain by

$$\langle F_+(\omega)\rangle = \frac{P_K N_K}{2} \frac{\frac{2qA_{He}}{N_{He}}\langle N_+(\omega)\rangle + g_I(q-1)\mathcal{A}_+(\omega)}{\omega_K - \omega - i\Gamma_K}. \quad (7)$$

Substitution of Eq. (3) in Eq. (7) directly yields Eq. (4), with the effective coupling coefficient reading

$$g_{eff} \equiv (q-1)\epsilon_P g_{aPP} - \xi g_{aNN}. \quad (8)$$

We can therefore identify the first unitless coefficient $\zeta = (q-1)\epsilon_P \approx 0.69$ and the second unitless coefficient as

$$\xi(\omega) = \frac{q\epsilon_N P_{He} A_{He}}{\omega - \omega_{He} + i\Gamma_{He}} \approx \frac{q\epsilon_N P_{He} A_{He}}{\omega}. \quad (9)$$

The last approximation pertains to the spectral content far from the helium resonance, because in our search range $\omega \gg \omega_{He}, \Gamma_{He}$ as $\omega$ is near $\omega_K$. We emphasize that, under this approximation, $\xi$ does not depend on $\Gamma_{He}$ or $\omega_{He}$. The reason is that the angle of the Helium spins is primarily modulated by the very large oscillation rate $\omega$, which corresponds to the rate at which the sign of the driving field changes. The effect of the finite coherence time of the helium spins or their exact resonance frequency in this driven configuration is relatively small.

### Background and signal sensitivity
The detector is sensitive to both real and anomalous magnetic fields. In this section, we present and characterize the noise model for the detector and analyze the sources of noise that limit its detection sensitivity in most frequencies. The dominant source of noise above 2 kHz, is magnetic field noise, as can be inferred from the noise being enhanced when measured at frequencies corresponding to the magnetic resonance. In the above regime, the polarization noise of the probe beam is likely the dominant non-magnetic noise source, as the noise at non-magnetically sensitive frequencies is consistent with the estimated photon shot noise. Below 2 kHz, (as well as at spectrally narrow noise-spikes), the origin of the noise cannot be reliably determined to be magnetic, or non-magnetic, due to the lack of measurements whose magnetic field is near resonance. See SI for further details.

The magnetometer signal is proportional to the collective spin component $\langle F_x\rangle = \text{Re}(\langle F_+\rangle)$ whose response to ALPs is given in Eq. (4). In the presence of noise, the collective spin measured by the detector reads as $\langle F_+\rangle \to \langle F_+\rangle + \delta F_+$ where the noise-driven contribution at frequency $\omega \gg \omega_{He}, \Gamma_{He}$ is given by

$$\delta F_+(\omega) = \left(\gamma_e - \frac{\xi(\omega)\gamma_{He}}{\epsilon_N}\right)\frac{P_K N_K}{2}\frac{\delta B_+(\omega)}{\omega_K - \omega - i\Gamma_K} + W(\omega). \quad (10)$$

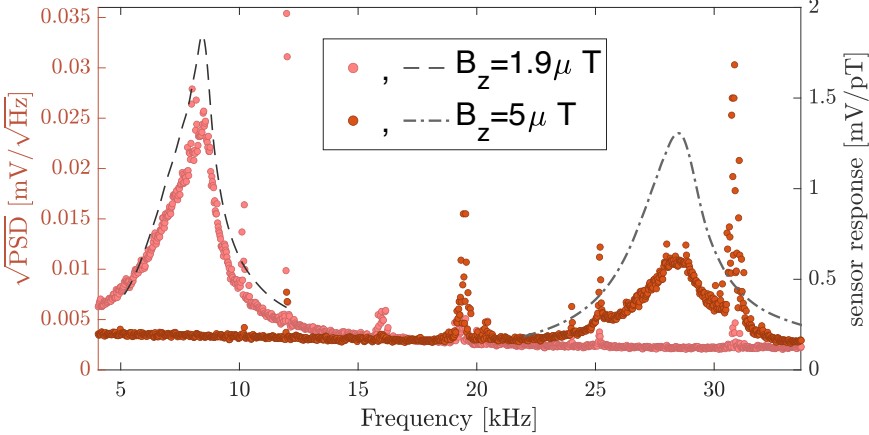

**Fig. 3 | Noise spectral density and magnetic response to external oscillating magnetic fields of two measurements.** Square root of the noise Power Spectral Density (PSD) of the detector (left axis) at two different magnetic fields in the search ($B_z = 1.9\mu$T, pink circles and $B_z = 5\mu$T, brown circles). Apart for some peaks, the spectrum generally follows an offset Lorentzian profile, owing to magnetic field noise and limited sensitivity of the detector. Dashed and dotted-dashed curves show the measured detector response to weakly oscillating transverse magnetic fields along the $y$ direction for $B_z = 1.9$, $5\mu$T respectively (right axis). The noise spectral densities and magnetic responses of additional configurations with different values of $B_z$ are provided in the Supplementary Information and in[57].

The first term describes the effect of magnetic field noise $\delta B_+ = \delta B_x + i\delta B_y$ at frequency $\omega$ when included in Eqs. ((5)–(6)) by $\mathbf{B} = \delta B_x \hat{\mathbf{x}} + \delta B_y \hat{\mathbf{y}} + B_z \hat{\mathbf{z}}$. It describes tilting of the total potassium spin that is driven by coupling of the electron spin to the magnetic field (with gyromagnetic ratio $\gamma_e = 2.8 \times 10^{10}$ Hz/T, first term in Eq. (10)), or to the transverse spin-exchange shift that is exerted by the tilted collective spin of the helium (with gyromagnetic-ratio $\gamma_{He} = 3.2 \times 10^7$ Hz/T, second term in Eq. (10)). The last term $W(\omega)$ denotes the technical noise originating primarily from measurement of the probe beam.

In Fig. 3 we exemplify the noise characteristics of the detector by presenting the square root of the power spectral density (PSD) of two recordings using Welch's method. The recordings were taken at two different magnetic fields $B_z = 1.9\mu$T (pink circles) and $B_z = 5\mu$T (brown circles) for 5 s each. The two noise spectra are presented in the raw units of the measuring device. The detector response to transverse oscillating magnetic-fields at these conditions are independently measured, and shown with dashed, and dotted-dashed lines respectively.

In Fig. 4, we present the total power spectral density of our detector, which is a combination of several different measurements. We used twenty-nine measurements, each lasting five seconds, taken at different values of $B_z$. The latter values generated different EPR resonances in the range of 4 – 42.5kHz. We tiled the spectrum by extracting the data at each frequency from the measurements whose magnetic response function is maximal. These measurements were scaled to units of magnetic field noise using the our independent magnetic calibration measurements. This corresponds to scaling of both the real magnetic (or spin-dependent) noise and the technical noise by the calibration scaling, to units of magnetic field noise. The plot was originally computed with 0.2Hz resolution, but was then smoothed using a running median of kernel size 20Hz to make the central value of the noise more visible. As shown in the figure, the noise spectrum is relatively white in the range of 2 – 30kHz, with an amplitude of about 5 – 8 fT/$\sqrt{\text{Hz}}$, except for some narrow spikes. At low frequencies, part of our noise increase is associated with contribution of non-magnetic noise, due to the weak magnetic response of our detector at these frequencies.

We also present the technical Noise Floor (NF) in blue. The NF is independently estimated using the off-resonance responses of two measurements, each with the EPR frequency near one limit of our scan range (each is used for the region where the other is closer to being in-

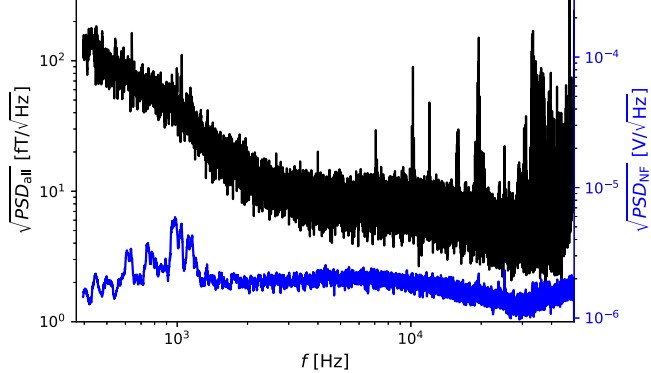

**Fig. 4 | Combined noise spectral density of the detector.** The black curve shows the square root of the total Power Spectral Density (PSD) of the detector, constructed by combining the magnetically sensitive range from twenty-nine separate measurements taken at different values of $B_z$ and different EPR frequencies. At each frequency, the noise is taken from the measurement with the largest response as measured by independent magnetic calibrations and properly converted to magnetic field units using said calibrations. The blue curve shows the technical Noise Floor (NF) in units of V/$\sqrt{\text{Hz}}$, estimated from the combined off-resonance spectrum of two measurements where the magnetic sensitivity is low. The technical noise level can be converted to units of magnetic noise, with a simple frequency-independent proportionality factor between the frequency range of about 2-45 kHz, wherein the left-hand vertical axis can also be used to read the NF. See the text for further details.

resonance). We find that the technical noise spectrum is relatively white with an amplitude of ~1 – 2$\mu$V/$\sqrt{\text{Hz}}$ for frequencies above 2 kHz. This noise level can be converted to an equivalent magnetic noise value by scaling through the maximum of our calibration measurement. In practice this maximum is changed by 34% between the two measurements used to make the plot (with most other measurements lying between these two extreme values). We have used the mean of the two responses at the peaks for the conversion shown in the figure. This scaling converts the technical noise floor to units of effective magnetic field with an amplitude of ~1 – 2 fT/$\sqrt{\text{Hz}}$. For frequencies $f \lesssim 2$ kHz or $f \gtrsim 45$ kHz, this scaling factor is invalid, owing to the lack of on-resonant measurements in said frequencies, which would lead to an increased contribution of the NF to the total noise in effective magnetic units.

The magnetic noise is consistent with the estimated noise produced by our innermost magnetic shield in the cell location[74], and is the limiting noise of our near-resonance measurement at most frequencies. The technical noise variance determines the realized sensitivity of our detector; It is consistent with shot noise of the power of the probe beam and is considered as spin-independent because it is almost unchanged by turn off of the pump beam.

## Calibrations

We have routinely calibrated the slowing down factor $q$ by measurement of the EPR frequency as a function of $B_z$. The slope of the linear fit yields the gyromagnetic ratio of the polarized potassium which corresponds to $\gamma_e/q$.

We estimated the spin-exchange field produced by the helium spins using the measured data as detailed in the data processing section. This estimation was validated by a measurement of the EPR the frequency during the spin-exchange optical-pumping stage (several hours process, preceded by application of strong magnetic fields that zeroed the initial helium polarization). We fitted it to the function

$$\omega_K(t) = \omega_K(0) + A_{He} P_{He}(1 - e^{-t/\tau}). \quad (11)$$

We calibrated the sensitivity of the magnetometer to weakly oscillating magnetic fields at the beginning and at the end of every measurement round, in between changes of the magnetic field value. We applied an additional calibration field $B = B_0 \cos(\omega_y t)\hat{y}$ with amplitudes $B_0 \leq 0.27$nT, and measured the response for several $\omega_y$ sampled around the EPR resonance. This measurement enabled the construction of the spectral form of the response function. The width and calibration factor remained very stable (<10% drift) during a single measurement. The EPR resonance however was decreased during the measurement because of temporal change in $P_{He}(t)$; While the helium spins were polarized at $B = 7\mu$T at the onset of each measurement, at lower magnetic fields the decoherence of the helium spins rate was moderately increased, primarily due to spatial inhomogeneity in the alkali-metal spins polarization[54]. The temporal variation in the spin-exchange field which shifted the EPR frequency was included in the analysis.

We would like to point out that precise calibrations of the linewidth and NMR frequency are not required for the computation of our bounds, assuming the approximation in Eq. (9). Instead, the quantity that is needed for the bounds and requires calibration is $A_{He}P_{He}$, which was estimated using the alkali-metal EPR frequency $\omega_K$, as explained in the data processing section. The fitting to Eq. (11) was used to validate the estimation of $A_{He}P_{He}$ but was not employed in the calculation of the bound itself. This is because the fitting only provides $A_{He}P_{He}$ at the start of the measurement and cannot be used to determine its time dependence.

## Data processing

We analyzed the data using the log-likelihood ratio test to exclude the presence of ALPs at frequency $\omega_{DM}$ with a width determined by the signal coherence time and the effects of earth's rotation on the sensitive axes of the detector. To accurately account for the velocity distribution of the Dark Matter, we carried the analysis procedure that is similar to ref. 14 except for a few changes listed below and further detailed in the SI. In short, we take the total ALP field as a superposition of the plane-waves representing the individual particles, whose momentum distribution follows the Standard Halo Model[44] and phases are uncorrelated and randomly distributed. To calculate the signal generated by the total ALP gradient, we compute the response of the detector in the frequency domain (which is represented by the $\alpha$ matrices as defined in ref. 14 with some minor changes discussed in the SI). Once the spectral content of an ALP has been determined, we use frequencies outside the predicted spectral content of the ALP to estimate the level of noise for the white noise hypothesis used to generate the bounds.

All analysis procedures and cuts were designed in a blind fashion, and decided in advance before looking at the data, to eliminate bias. We emphasize that this is an exclusion search, and no discovery was attempted. We carried a similar procedure for the quality cuts as described in ref. 14, primarily to avoid a change in the analysis procedure, which presumably mitigated low-frequency noise; Indeed the noise spectrum of the unblinded data was nearly unchanged. We also note that owing to the long measurement time, spectral representation of the ALP required fewer points that in ref. 14, rendering its computation more efficient.

The magnetization of the helium has decayed during the measurement, owing to its magnetic-field dependent lifetime. This rendered both $\xi$ and $\omega_K$ time-dependent, slowly-varying throughout the measurement at the scale of many minutes. Because this time scale is much longer than $1/\omega_{DM}$, we assumed that $P_{He}$, and $\omega_K$ change adiabatically, such that Eqs. ((3)-(7)) remain valid at short time scales. We first classified if a decay is negligible based on the following criterion: we compared the EPR frequencies of the magnetic-response function measured at the calibration preceding and appending the recording and checked for variation that is >200 Hz. This criterion classified most runs with $f_K(B_z) \gtrsim 20$kHz as cases with negligible decay, where we took the average of the two EPR frequencies and the average of helium polarizations as constant values (note that in this regime $\Gamma_K \gtrsim 800$ Hz). For the remaining cases, we estimated the time-dependent function $\omega_K$. To do this, we computed the spectrum of the data in a window of approximately one second every few seconds and fit it to a combination of response functions obtained during the calibration stages. The central frequency of the fitting function was taken as $\omega_K(t)$. Because the fitting function is much wider in frequency than ALP signals, this procedure does not introduce any bias to the search. In the adiabatic approximation, at any given moment, $A_{He}P_{He}(t) = \omega_K(t) - B_z\gamma_e/q$, which is how $A_{He}P_{He}$ was estimated using the above prescription to find $\omega_K(t)$. The time-dependence of the response was then used to compute $\alpha$ (see[14] and the Supplementary Information for the slight modifications from[14]).

## Data availability

A repository of all processed data needed to redo the statistical analysis is available in two parts in refs. 80,81. The second dataset contains a readme file which elaborates on the contents of the other files. The two datasets contain the Fourier transform of the original measurements, after the quality cuts were made, and filtering was done, and only in frequencies where they had a meaningful sensitivity to the signal. The datasets also contain the ALP covariance matrices, and the matrices used to transition from the ALP covariance matrices to the contribution of the ALPs to the detector. All of the above files (except for the readme file) are .pkl files, which may be read by importing the pickle package of python. In addition, a folder with all calibration measurements as matlab files is within the repository. The constraints on ALP-neutron and ALP-proton interactions at all individual masses computed can be found in ref. 57 (as .pkl files). This repository also contains a readme with explanation on the contents of the folder (as a text file), as well as additional calibration measurements, and spectra at frequencies with a significant statistical excess, as discussed in the SI (as pdf files).

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

## Acknowledgements

The work of Y.H. is supported by the Israel Science Foundation (grants No. 1112/17 and 1818/22), by the Binational Science Foundation (grant No. 2016155), by the Azrieli Foundation and by an ERC STG grant ('Light-Dark', grant No. 101040019). E.K. is supported by the US-Israeli Binational Science Foundation (grant No. 2020220) and by the Israel Science Foundation (grant No. 1111/17). T.V. is supported by the Israel Science Foundation (grant No. 1862/21), by the Binational Science Foundation (grant No. 2020220) and by the European Research Council (ERC) under the EU Horizon 2020 Program (ERC-CoG-2015—Proposal n. 682676 LDMThExp). This project has received funding from the European Research Council (ERC) under the European Union's Horizon Europe research and innovation program (grant agreement No. 101040019). Views and opinions expressed are however those of the author(s) only and do not necessarily reflect those of the European Union. The European Union cannot be held responsible for them.

## Author contributions

Authors I.M.B., R.S., Y.H., E.K., T.V. and O.K. contributed to the experimental design, construction, data collection, and analysis of this experiment.

## Competing interests

The authors declare no competing interests.
