## [Peer Review File · Nature Communications]

Constraints on axion-like dark matter from a SERF comagnetometerREVIEWER COMMENTS

Reviewer #1 (Remarks to the Author):

The manuscript reports on novel experimental results using a comagnetometer to search for axion-like-particles (ALP) that couple to protons or neutrons. They obtain the strongest terrestrial constraints to date over a sizeable range of ALP masses, in particular extending previous searches to much higher masses. Such searches for ALPs are a crucial component of the hunt for dark matter. Although I think it is fair to say that the experiment has not yet reached the most interesting parts of parameter space (see comments below), this manuscript represents an important step forward. The results are noteworthy and significant enough both for the field of axion detection, as well as dark matter physics in general, to merit publication in nature. I cannot comment on the details of the experimental setup, but the methods are well explained for a non-expert.

I do however have two comments (neither of which should prevent the manuscript being published), of which I think the first is important to clarify, while the authors may want to consider the second but this is not essential:

1. The authors claim that neutron star cooling does not lead to constraints on the axion-proton coupling. However, the references they give suggest otherwise. E.g. the manuscript's reference [50] claims that $\sim 30\text{-}40\%$ of a proto-neutron star consists of protons and gives a limit on the combined coupling $g_{\text{ann}}^2 + g_{\text{app}}^2$, and reference [53] states "For the value of $C_n=0.04$ we adopted and for our choice of pairing gaps, the axion emission is dominated by proton condensate". The authors should either explain why they do not think the coupling to protons is constrained, or adjust the manuscript and Figures to reflect these bounds.

2. For typical axion couplings to protons and neutrons (roughly of size $\sim 1/f_a$) it is not at all straightforward to get the full dark matter abundance in the part of parameter space that the experiment is currently probing (see for instance Figure 1 of <https://arxiv.org/abs/1201.5902> for the situation in the pre-inflationary scenario and the situation is similar in the post-inflationary scenario). Even, e.g. modifying the cosmological

history prior to big bang nucleosynthesis or considering more complicated theories, this does not appear sufficient. The article is of course primarily experimental and the results remain important progress towards the most motivated parts of parameter space, but perhaps a comment would be useful for readers from outside the field of axion cosmology.

Reviewer #2 (Remarks to the Author):

In the manuscript “New constraints on axion-like dark matter from a SERF comagnetometer” I.M. Bloch et al. describe the result of a ~1 month long search for axion-like particle (ALP) dark matter with coupling to protons and neutrons in the ALP mass range of 1.4-200peV (corresponding to 0.33-50 kHz ALP Compton frequency) using a 3He-K comagnetometer. The cast bounds by this work are astronomically excluded but nonetheless very relevant to the physics community as there are no other experimental searches in this parameter range.

However, as the authors put it, they do not attempt at a possible discovery. While this is a confusing statement for a dark matter search paper, it is probably owed to the fact that the data is littered with not commented on peaks that go off the exclusion plot scale. The analysis is probably sound and, even though the peak density seems enormous from the figure, on average (ignoring the potential candidates) this exclusion is the strongest in this mass range. But the complete lack of addressing this experimental flaw is the biggest weakness of the paper. Other groups go great length to analyze their noise sources and try to argue why certain dark matter candidates cannot be dark matter but noise, which is the appropriate approach in our view, but I.M. Bloch et al. present their results as is, including all the peaks and the physics community can check the actual exclusion limit for a certain frequency in the published data set or, formulated differently, find arguments themselves why certain oscillating magnetic fields are not dark matter but noise. This effectively renders the data useless, because it is probably easier to scan this whole range again (with a less noisy experiment) instead of trying to rescan the myriad of peaks/potential ALP candidates. The authors also remain unsatisfactory vague in several important technicalities of their work. (quote:” We varied the magnetic field at few tens of discrete points...” How many points is important, which magnetic fields is important, too) There is no proper noise floor of the experiment over the complete exclusion region (330-50000Hz). They claim

2fT/Hz^{1/2} sensitivity of the K magnetometer and a 10fT/Hz^{1/2} “nearly white” magnetic noise floor. Both quantities are impressive and warrant a demonstration in the methods or supplemental material. Especially considering that they apply a 7E-6 magnetic field with an electromagnet that then requires a fractional stability of 1ppb. Which is even more impressive. Dark matter search papers are mostly about noise floors, so it is worth displaying them. In this manuscript is effectively no noise analysis.

And there is no data on the calibration/sensitivity of the Helium. As far as we understand, they look at the very off-resonant response of the nuclear helium spin (in the neutron coupling plot) to fictitious or real oscillating magnetic fields. For 70mG the Helium nuclear magnetic resonance is at 220Hz, so the response to higher frequencies is suppressed. But the suppression scales with the linewidth of the nuclear magnetic resonance and the authors just give an upper limit of 0.1 Hz to this quantity which is most likely strongly and nonlinearly dependent on the applied Bz field. It can also be technically much smaller than 0.1Hz, which would strongly affect their exclusion limits. Omission of the measured ³He NMR feature cast doubts on their limits. Also the EPR resonance calibration is missing from the paper. Two resonances are shown, that look vastly different. Why not show all?

Reviewer #3 (Remarks to the Author):

The paper reports on new bounds on ultralight axion-like-particle (ALP) dark matter via its potential coupling with protons and neutrons. The technique is based on a SERF comagnetometer that uses a mixture of noble-gas and alkali-metal atoms. Varying the applied magnetic field the precession resonance can be tuned and thus the sensitivity to different axion masses. In this way, bounds on both ALP-proton and ALP-neutron couplings for ALP masses in the range 1.4 to 200 peV are produced. These bounds improve previous experimental limits on these couplings. The sensitivity is very far from the expectations from QCD axion models. Moreover, the derived bounds are still in the region that is disfavoured by astrophysics considerations involving the observation of supernova SN1987A. However, as the authors point out, the uncertainty of such considerations is substantial and should not discourage experiments from probing this region. On the technical side, the possibility of simultaneously probing neutron and proton coupling by using the dual atomic species is a remarkable innovation. In summary, I consider this a valuable result in the search for ALP

dark matter that deserves publication. I have more doubts on whether the interest of the results outside this specific field is of sufficient entity so as to warrant publication in a journal like Nat. Comm.

REVIEWER #1:

The manuscript reports on novel experimental results using a comagnetometer to search for axion-like-particles (ALP) that couple to protons or neutrons. They obtain the strongest terrestrial constraints to date over a sizeable range of ALP masses, in particular extending previous searches to much higher masses. Such searches for ALPs are a crucial component of the hunt for dark matter. Although I think it is fair to say that the experiment has not yet reached the most interesting parts of parameter space (see comments below), this manuscript represents an important step forward. The results are noteworthy and significant enough both for the field of axion detection, as well as dark matter physics in general, to merit publication in nature. I cannot comment on the details of the experimental setup, but the methods are well explained for a non-expert.

I do however have two comments (neither of which should prevent the manuscript being published), of which I think the first is important to clarify, while the authors may want to consider the second but this is not essential:

1. The authors claim that neutron star cooling does not lead to constraints on the axion-proton coupling. However, the references they give suggest otherwise. E.g. the manuscript's reference [50] claims that $\sim 30 - 40\%$ of a proto-neutron star consists of protons and gives a limit on the combined coupling $g_{nn}^2 + g_{pp}^2$, and reference [53] states "For the value of $C_n = 0.04$ we adopted and for our choice of pairing gaps, the axion emission is dominated by proton condensate". The authors should either explain why they do not think the coupling to protons is constrained, or adjust the manuscript and Figures to reflect these bounds.

We thank the Reviewer for this comment, and completely agree that the proto-neutron star sets constraints on both the neutron-axion (g_{nn}) and proton-axion (g_{pp}) couplings. Like many stellar cooling constraints, the range of excluded coupling strengths is bounded from above. That is because coupling constants that are too large result in the ALPs becoming trapped in the core of the star, limiting the emission rate. The numerical values of the excluded neutron-axion and proton-axion couplings are different, owing to their different mean free paths. We adapt the analysis of Ref. [PRD 98 103015 (2018)] and estimate that the axion-proton constraint is upper bounded by $g_{pp} < \sim 10^{-6} \text{GeV}^{-1}$, which is lower than the limits of our constraint on the coupling to protons. We did not show this in the previous version of the manuscript as it lay outside the plotted region of the figure.

To clarify this point, in the revised manuscript we increased the plotted range in Fig.2b to include the axion-proton bounds set by the proto-neutron star (NS). We present this constraint in the caption of Fig. 2, and in the main text, and further discuss its origin and the adapted assumptions in the Search Results section.

2. For typical axion couplings to protons and neutrons (roughly of size $\sim 1/f_a$) it is not at all straightforward to get the full dark matter abundance in the part of parameter space that the experiment is currently probing (see for instance Figure 1 of <https://arxiv.org/abs/1201.5902> for the situation in the pre-inflationary scenario and the situation is similar in the post-inflationary scenario). Even, e.g. modifying the cosmological history prior to big bang nucleosynthesis or considering more complicated theories, this does not appear sufficient. The article is of course primarily experimental and the results remain important progress towards the most motivated parts of parameter space, but perhaps a comment would be useful for readers from outside the field of axion cosmology

Agreed. We have added the following comment in the discussion section:

“Various cosmological scenarios can give a variety of ALP couplings that would produce the observed cold dark matter density [JCAP 06 013 (2012)]. However, the range of ALP-neutron or ALP-proton couplings predicted are typically several orders of magnitude smaller than our current bounds. Nevertheless, sensors based on ensembles of alkali-metal and noble-gas spins have the potential to explore this theoretically-motivated region in the ALP parameter space, thanks to their long coherence time and the size of the ensemble [Quant. Sci. Tech 6 3 (2021)]. In addition to setting new constraints, our work represents progress towards experimental exploration of this theoretically-motivated regime by leveraging the high sensitivity of SERF sensors for this frequency range to search for new physics.

Several techniques are expected to improve the performance of our sensor...”

We then follow this comment by describing several techniques that can improve the performance for the future generation of the detector.

REVIEWER #2:

In the manuscript “New constraints on axion-like dark matter from a SERF comagnetometer” I.M. Bloch et al. describe the result of a ~1 month long search for axion-like particle (ALP) dark matter with coupling to protons and neutrons in the ALP mass range of 1.4-200peV (corresponding to 0.33-50 kHz ALP Compton frequency) using a $^3\text{He-K}$ comagnetometer. The cast bounds by this work are astronomically excluded but nonetheless very relevant to the physics community as there are no other experimental searches in this parameter range.

However, as the authors put it, they do not attempt at a possible discovery. While this is a confusing statement for a dark matter search paper, it is probably owed to the fact that the data is littered with not commented on peaks that go off the exclusion plot scale. The analysis is probably sound and, even though the peak density seems enormous from the figure, on average (ignoring the potential candidates) this exclusion is the strongest in this mass range. But the complete lack of addressing this experimental flaw is the biggest weakness of the paper. Other groups go great length to analyze their noise sources and try to argue why certain dark matter candidates cannot be dark matter but noise, which is the appropriate approach in our view, but I.M. Bloch et al. present their results as is, including all the peaks and the physics community can check the actual exclusion limit for a certain frequency in the published data set or, formulated differently, find arguments themselves why certain oscillating magnetic fields are not dark matter but noise. This effectively renders the data useless, because it is probably easier to scan this whole range again (with a less noisy experiment) instead of trying to rescan the myriad of peaks/potential ALP candidates.

We thank the Reviewer for this comment, and respond to this apt point by considerably extending our analysis and providing further details on the measured peaks. In our frequency range, there are about 5×10^6 independent ALP candidates, as can be estimated by binning the studied frequency range with a typical ALP bandwidth of about $\sim 10^{-6} f_a$. Among these candidates, we find that only about 3788 spectral points are inconsistent with our white-noise model (i.e. appear as small or large peaks in the spectrum), and therefore could be ALP candidates. To study these peaks, we first use available experimental data (that was previously designated to study our quality cuts but was unused) and recompute the constraints. This data was taken at a different time with respect to the rest of the experiment. The constraints are found to improve at several regions, and the number of candidates that are inconsistent with our white-noise model reduced to 1240. The persistence of these candidates indicates that the source which generates them is relatively stationary, and taking further data with our apparatus at present is unlikely to improve these results.

Next, we show that the spectral properties of the observed peaks are inconsistent with those expected for ALPs. We find that the great majority of peaks have a considerably smaller bandwidth compared to that of the ALP. We introduce an additional test that excludes the narrow peaks, reducing the number of candidates to only 5, among the entire spectrum, with only 2 of those 5 not having other peaks very near them (which would suggest they are sourced by technical noise, as ALP signal has no reason to be so near other peaks which are presumably just noise). This analysis is now presented in the main text, and detailed in the Supplementary Information. We note that while this analysis stage could have improved the bounds at the location of the peaks, we do not change the reported values in Fig. 2 owing to this process being conducted post-unblinding.

As a short note, the appearance of spurious peaks that cannot be completely excluded, is a common theme in long searches with high resolution and statistical sensitivity. As an example, both [PRL 129 031301 (2022)] and [PRL 127 081801 (2021)] observed a similar or greater number of unexcluded peaks (before unblinding) that are inconsistent with the null hypothesis of their noise model. We therefore believe that the data is not useless, but on the contrary — it provides constraints on the great majority of the spectrum up to a few points. We agree, however, that an introduction of new techniques that can discriminate noise from ALPs (and thus providing detection capability) could elucidate valuable information on the noise. In the discussion section we now introduce and discuss two different techniques that can potentially provide this capability.

We would also like to highlight that the seemingly “high density of peaks” in our results partially originated from the visual way that the bounds were presented in Fig. 2. Due to the finite pixel size of monitors and printers, it is technically impossible to put the entire information of the bounds in our range that in principle consists of 5×10^6 different spectral points (the full information on our bound is provided in an online data repository at github). For that purpose, to compactify the amount of presented information in the figure, we bin different regions in the spectrum (on a logarithmic scale), and plot the mean, standard deviation and minimal and maximal values within these bins. In the previous version, we had used only about 500 bins (binning masses logarithmically with 1% bins) for the entire frequency range, which resulted in a large density of plotted peaks. But in fact, the peaks only take 0.08% of the scanned spectrum. In the revised version, we use around 50,000 bins instead (binning masses logarithmically with 0.01% bins), better conveying that peaks are present but with a clearer view of their extent. We have updated our text and caption to include the binning information.

The authors also remain unsatisfactory vague in several important technicalities of their work. (quote:” We varied the magnetic field at few tens of discrete points...” How many points is important, which magnetic fields is important, too)

Agreed. We now give more details on the values of the magnetic field scanned throughout the search in Table I in the Supplementary Information, and in the github repository we present the full set of points.

There is no proper noise floor of the experiment over the complete exclusion region (330-50000Hz). They claim $2\text{fT}/\text{Hz}^{1/2}$ sensitivity of the K magnetometer and a $10\text{fT}/\text{Hz}^{1/2}$ “nearly white” magnetic noise floor. Both quantities are impressive and warrant a demonstration in the methods or supplemental material. Especially considering that they apply a $7\text{E}-6$ magnetic field with an electromagnet that then requires a fractional stability of 1ppb. Which is even more impressive. Dark matter search papers are mostly about noise floors, so it is worth displaying them. In this manuscript is effectively no noise analysis.

We thank the reviewer for raising this important point, and respond by providing more detail about our noise analysis. We now present the composite magnetic noise spectrum of our entire search in Fig. 4 and in the Methods section, which is post-processed by combining the noise spectrum of individual measurements near the frequencies at which the magnetic sensitivity is high. As can be seen, aside from some spurious narrow peaks, the noise is relatively white in the frequency range of 2-30 kHz, with amplitudes ranging between $(5 - 9) \text{fT}/\sqrt{\text{Hz}}$ designating the *total* noise floor. We further analyze the detector sensitivity,, by considering the off resonant noise spectrum combined by two measurements. We show that this noise floor, which can originate from shot noise of our probe beam or electric noise at the detector, is at the level of about $2 \text{fT}/\sqrt{\text{Hz}}$ at most of the analyzed frequency range. In the SI, we further show that the spectral shape of our noise power spectral density at each measurement is consistent with the independently measured magnetic response function, therefore highlighting that the noise source is likely magnetic.

We would like to comment that the magnetic noise at which the detector is sensitive to, is oriented in the x-y plane and oscillates at the frequencies considered in the scan range. The bias magnetic field that controls the Larmor frequencies on the other hand, is oriented along z and is at DC (constant in time). Thus, the magnetic noise is probably not limited by the stability of the magnetic field value. We estimate that it is consistent with Johnson noise generated by our inner magnetic shield.

And there is no data on the calibration/sensitivity of the Helium. As far as we understand, they look at the very off-resonant response of the nuclear helium spin (in the neutron coupling plot) to fictitious or real oscillating magnetic fields. For 70mG the Helium nuclear magnetic resonance is at 220Hz, so the response to higher frequencies is suppressed. But the suppression scales with the linewidth of the nuclear magnetic resonance and the authors just give an upper limit of 0.1 Hz to this quantity which is most likely strongly and nonlinearly dependent on the applied B_z field. It can also be technically much smaller than 0.1Hz, which would strongly affect their exclusion limits. Omission of the measured ^3He NMR feature cast doubts on their limits.

We thank the Reviewer for this question, and apologize for the confusion. The neutron-ALP exclusion plots that we generate in this work have very small dependence on Γ_{He} or ω_{He} . This is because the exact values of Γ_{He} and ω_{He} affect our constraints only through the unitless coefficient ξ , given by Eq. (9). Because our operation is far off resonant (we study frequencies at which $\omega \gg \omega_{\text{He}}, \Gamma_{\text{He}}$), the spin response of the Helium is moderated by the large frequency of the drive and not by the linewidth of the NMR spectra, as the right hand side of Eq. (9) entails. The intuitive picture is that the helium spins respond to a drive field whose sign oscillates at a rate ω . Owing to this rapid sign change, the angle of the Helium would also be oscillatory, and its amplitude would correspond to the strength of the field times a fraction of the oscillation time (and would be independent of the coherence time, even if the latter would have been infinitely long). We now present this intuitive explanation in the text below Eq. (9) and emphasize the very small dependence of ξ on Γ_{He} or ω_{He} . We also provide more details on the estimation of $A_{\text{He}} P_{\text{He}}(t)$, the strength of the Helium magnetization as sensed by the potassium, in the Supplementary Information.

Also the EPR resonance calibration is missing from the paper. Two resonances are shown, that look vastly different. Why not show all?

We thank the Reviewer for this suggestion. The experiments consisted of 110 different values of the bias magnetic field, each with a different calibration signal. We now present a dozen other calibration signals in the Supplementary Information and additionally uploaded to the data repository (github) all measured calibrations.

Reviewer #3:

The paper reports on new bounds on ultralight axion-like-particle (ALP) dark matter via its potential coupling with protons and neutrons. The technique is based on a SERF comagnetometer that uses a mixture of noble-gas and alkali-metal atoms. Varying the applied magnetic field the precession resonance can be tuned and thus the sensitivity to different axion masses. In this way, bounds on both ALP-proton and ALP-neutron couplings for ALP masses in the range 1.4 to 200 peV are produced. These bounds improve previous experimental limits on these couplings. The sensitivity is very far from the expectations from QCD axion models. Moreover, the derived bounds are still in the region that is disfavoured by astrophysics considerations involving the observation of supernova SN1987A. However, as the authors point out, the uncertainty of such considerations is substantial and should not discourage experiments from probing this region. On the technical side, the possibility of simultaneously probing neutron and proton coupling by using the dual atomic species is a remarkable innovation. In summary, I consider this a valuable result in the search for ALP dark matter that deserves publication. I have more doubts on whether the interest of the results outside this specific field is of sufficient entity so as to warrant publication in a journal like Nat. Comm.

We thank the Reviewer for their comments. We have added in the revised text a comment that emphasizes that our constraints are yet far from the targets provided by most models wherein the ALPs can be produced sufficiently to explain the observed DM abundance, which is a closer target than the QCD axion line. We also further discuss future routes to improve the sensitivity of the apparatus in future searches:

“Various cosmological scenarios can give a variety of ALP couplings that would produce the observed cold dark matter density [JCAP 06 013 (2012)]. However, the range of ALP-neutron or ALP-proton couplings predicted are typically several orders of magnitude smaller than our current bounds. Nevertheless, sensors based on ensembles of alkali-metal and noble-gas spins have the potential to explore this theoretically-motivated region in the ALP parameter space, thanks to their long coherence time and the size of the ensemble [Quant. Sci. Tech 6 3 (2021)]. In addition to setting new constraints, our work represents progress towards experimental exploration of this theoretically-motivated regime by leveraging the high sensitivity of SERF sensors for this frequency range to search for new physics. Several techniques are expected to improve the performance of our sensor...”

We thank the three Reviewers for their excellent suggestions.

REVIEWERS' COMMENTS

Reviewer #1 (Remarks to the Author):

In reply to my comment 1, the authors have added a discussion claiming that the values of $g_{\{aPP\}}$ that their experiment is sensitive to are allowed due to trapping effects in neutron stars and modified the plot accordingly.

I find it somewhat implausible that the effects of trapping are so different for the ALP proton coupling compared to that to neutrons: It is not at all clear to me why the upper limit on the bound on $g_{\{aPP\}}$ should be a factor ~ 100 smaller than that on $g_{\{aNN\}}$ and no explanation is given beyond citing two different references. Nevertheless, it is true that limits from cooling are uncertain in this regime (in reality, probably both for $g_{\{aPP\}}$ and $g_{\{aNN\}}$). I would prefer more caution in this discussion in the manuscript but I do not insist.

If the experimental aspects are sufficiently novel (which the other referees appear better positioned to judge), I think the manuscript can be published in nature.

Reviewer #2 (Remarks to the Author):

In their revision I.M. Bloch et al. addressed all our raised objections. The manuscript is highly relevant for a wide range of physics communities and represent the strongest bounds on bosonic dark matter coupling to neutron or proton dominated nuclear spins. A caveat remains the quality of the data: numerous peaks litter the spectrum, and even though the authors address this issue appropriately, it reduces the impact of this work. It would maybe help to increase the bin width in Figure 2. 0.01% is not displayable in this format (corresponding to 10000 points in a maybe 3-inch-wide plot). Something like 0.1-0.2% would allow to visualize all data points in the plot. Overall, the manuscript would benefit (still) of more experimental detail, for example, the magnetic sensitivity calibration relies entirely on the written statement of the authors. The claimed sensitivity is impressive, in ranges better than Savukov et al, Phys. Rev. Lett. 95, 063004 (2005), which uses a bigger cell, at a higher temperature with a factor 8 smaller linewidth. Also, key experimental settings like time series duration or frequency resolution of the science data are missing.

And while the sharing of all data is commendable, I am unfortunately not aware of the software to look at it.

Some loose points:

-At the end of the method section, the author basically claim, that their noise floor is explained by magnetic shield noise. I do not believe this. Especially the $1/f$ up to 2 kHz seems at odds with this statement. Adding experimental information would also aid the statement, that the technical noise floor is shot-noise limited (laser beam power).

-The y-axis unit $\text{mV}/(\text{pT})^{1/2}$ is probably a typo in figure S1. (The top row of plots is far from Lorentzian btw)

-In Fig. S2: is the y-axis correct? The inset suggests the peak height in magnetic field units, however the y-axis is Voltage.

-the author should agree on a magnetic field unit and stick to it. Either G or T.

-“Narrow spikes have been removed by a set of filters” in the caption of S2. Why so vague? What kind of filters? Why are there still narrow peaks in the plots?

-The acronym PLR is not defined.

-The noise floor in S3 and S4 besides the peak varies by orders of magnitude. Why is that?

Reviewer #1 (Remarks to the Author):

In reply to my comment 1, the authors have added a discussion claiming that the values of g_{aPP} that their experiment is sensitive to are allowed due to trapping effects in neutron stars and modified the plot accordingly.

I find it somewhat implausible that the effects of trapping are so different for the ALP proton coupling compared to that to neutrons: It is not at all clear to me why the upper limit on the bound on g_{aPP} should be a factor~100 smaller than that on g_{aNN} and no explanation is given beyond citing two different references. Nevertheless, it is true that limits from cooling are uncertain in this regime (in reality, probably both for g_{aPP} and g_{aNN}). I would prefer more caution in this discussion in the manuscript but I do not insist.

We thank the Reviewer for this comment. The two different limits, calculated and reported by other independent groups, are only cited by us. While it is beyond the scope of this experimental work to analyze these theoretical estimates in detail, or to prefer one constraint over the other, we follow the suggestion and inform the reader about the possible discrepancy in the discussion section:

“We note that the two upper edges of these constraints are rough estimates that were calculated independently by other groups \cite{Beznogov:2018fda,Hamaguchi_2018} and carry a relatively large difference that may be an artifact of different approximations used, rather than some underlying difference in the physics..”

If the experimental aspects are sufficiently novel (which the other referees appear better positioned to judge), I think the manuscript can be published in nature.

We thank the Reviewer for this recommendation.

Reviewer #2 (Remarks to the Author):

In their revision I.M. Bloch et al. addressed all our raised objections. The manuscript is highly relevant for a wide range of physics communities and represent the strongest bounds on bosonic dark matter coupling to neutron or proton dominated nuclear spins. A caveat remains the quality of the data: numerous peaks litter the spectrum, and even though the authors address this issue appropriately, it reduces the impact of this work. It would maybe help to increase the bin width in Figure 2. 0.01% is not displayable in this format (corresponding to 10000 points in a maybe 3-inch-wide plot). Something like 0.1-0.2% would allow to visualize all data points in the plot.

Done. We have modified our resolution as per the request of the reviewer.

Overall, the manuscript would benefit (still) of more experimental detail, for example, the magnetic sensitivity calibration relies entirely on the written statement of the authors.

We now include more experimental details, and added additional information in the first paragraph of the apparatus subsection (see our answers below).

Regarding the magnetic calibration procedure, we have included our raw calibration measurements in the publicly available Zenodo database and added an explanation of these files in the Zenodo folder description.

The claimed sensitivity is impressive, in ranges better than Savukov et al, Phys. Rev. Lett. 95, 063004 (2005), which uses a bigger cell, at a higher temperature with a factor 8 smaller linewidth.

We thank the reviewer for acknowledging our magnetic sensitivity. Regarding the comparison with Savukov et al. (Phys. Rev. Lett. 95, 063004, 2005), we would like to note that they report an optical noise level around $0.7\text{fT}/\sqrt{\text{Hz}}$, as shown in figure 3. Therefore the two detectors are generally on par (yet their detector slightly outperforms our detector in terms of magnetic sensitivity).

Furthermore, we wish to correct some of the details of the comparison between the two experimental configurations. We estimate that they have approximately 4×10^{14} potassium spins, while we have around 5×10^{14} spins (as mentioned in the apparatus section). As their cell is larger, (4cm^3 to compared with our 1.4 cm^3 cell) our density (and hence temperature) is higher. Please note that Phys. Rev. Lett. 95, 063004, 2005 does mention 200cm^3 cells in the abstract, but these are not the ones in which the noise levels are reported. We have additionally added the explicit size of our cell diameter (1.4 cm) to the apparatus section, as it was previously only present in a reference.

Also, key experimental settings like time series duration or frequency resolution of the science data are missing. And while the sharing of all data is commendable, I am unfortunately not aware of the software to look at it.

We've added a detailed explanation to the Zenodo readme description (the .pkl files are opened by importing the pickle package of python). While all frequency points are available in the github, we included a table with additional information for several indicative measurements in Supplementary table S2.

Some loose points:

-At the end of the method section, the author basically claim, that their noise floor is explained by magnetic shield noise. I do not believe this. Especially the $1/f$ up to 2 kHz seems at odds with this statement. Adding experimental information would also aid the statement, that the technical noise floor is shot-noise limited (laser beam power).

We thank the reviewer for this comment. To clarify this matter we have added the following statement to the main text:

“Below 2 kHz, a reduced magnetic sensitivity and the lack of near resonant magnetic measurement hinder the decomposition of the noise into its magnetic and non-magnetic contributions (see Methods and SI). “

And the following paragraph to the Methods section:

“The dominant source of noise above 2 kHz, is magnetic field noise, as can be inferred from the noise being enhanced when measured at frequencies corresponding to the magnetic resonance. In the above regime, the polarization noise of the probe beam is likely the dominant non-magnetic noise source, as the noise at non-magnetically sensitive frequencies is consistent with the estimated photon shot noise. Below 2 kHz, (as well as at spectrally narrow noise-spikes), the origin of the noise cannot be reliably determined to be magnetic, or non-magnetic, due to the lack of measurements whose magnetic field is near resonance. See SI for further details.”

Regarding the laser power, we have added the power of the probe laser at the photodiodes, as well as the specific photodiodes used to the apparatus section:

“... using a pair of photo-diodes (Thorlabs PDB210A) in a homodyne configuration [48], with each of the two photo-diodes receiving about 0.5 mW of power (estimated using the Faraday rotation of the polarization during calibration measurements)”.

We note that the theoretically calculated photon shot noise floor of $PSN \sim 1.5 \mu V / \sqrt{\text{Hz}}$ [estimated as $\sqrt{2P_{\text{probe}} \cdot h \cdot \nu} / GR$, where $G = 175 \cdot 1000 \cdot V/A$, $R = 0.55 A/W$ (properties of PDB210A), and $\nu = 770 \text{ nm}$] is consistent with the measured one.

-The y-axis unit $\text{mV}/(\text{pT})^{1/2}$ is probably a typo in figure S1. (The top row of plots is far from Lorentzian btw)

Corrected. We also agree that the top row is not very well described by a Lorentzian, and added the following paragraph to section S1:

“As can be seen in Fig. S1, the response at higher frequencies ($\gtrsim 5 \text{ kHz}$) follows a Lorentzian profile as one would expect from a direct measurement of the left hand side of Eq. (4). However, at low frequencies, frequency dependence of the electrical filter used for the measurement, as well as interference from the Lorentzian centered around negative frequencies, lead to deviations from the Lorentzian profile. Our calibrations account for these deviations, with uncertainty of about $\sim 10\%$ at lower frequencies ($\lesssim \text{kHz}$), estimated by comparing the response at a given frequency to the one predicted by a linear fit to its neighboring measured responses. Pre-calibration measurements at low fields ($\sim 1.2 \mu T$), have an additional deviation from the Lorentzian profile, due to a small but non-negligible decay of the helium during the measurement. By comparing pre- and post-calibrations with the same resonance frequency, we may see that this causes up to $\sim 20\%$ change in the fitted Γ . Since the measurements at low magnetic fields are much

longer than the T_1 of the helium, the response is dominated by the response after the helium reaches its steady state. And, since $f_{\text{pre}}-f_{\text{post}} \sim \Gamma$ this has an altogether a minor effect.”

-In Fig. S2: is the y-axis correct? The inset suggests the peak height in magnetic field units, however the y-axis is Voltage.

The Y axis is correct. We now include this information in the caption:

“The label shows the value of the magnetic noise contribution to the measured noise at the peak in units of magnetic field noise.”

-the author should agree on a magnetic field unit and stick to it. Either G or T.

Done. We have amended all uses of G to T.

-“Narrow spikes have been removed by a set of filters” in the caption of S2. Why so vague? What kind of filters? Why are there still narrow peaks in the plots?

We now add the exact description of the new filters used:

“... we use a 2-Hz wide running median filter on the full noise spectrum, and a 0.2-Hz wide running median filter on the technical noise offset.”

Most remaining peaks are either quite wide, or come from a group of many nearby peaks, which are therefore sometimes unified as one wide fit under the filtering procedure.

-The acronym PLR is not defined.

Corrected. We now use the word Likelihood instead of profile-likelihood ratio (PLR).

-The noise floor in S3 and S4 besides the peak varies by orders of magnitude. Why is that?

These two figures are normalized to the maximal PSD (and therefore both are given in A.U.). We now explain this in the figure captions:

Fig S3: *“...We have normalized the peak of the ALP-signal, and the maximal $\sqrt{\text{PSD}}$ point in each figure to unity.”*

Fig S4: *“...Each noise spectrum was normalized such that its maximal $\sqrt{\text{PSD}}$ point is at unity, while the normalization of the ALP is determined from the fit to the (normalized) PSD.”*

We thank the reviewer for the thorough review of our manuscript.